# *N*-Glycosylation of TREK-1/hK_2P_2.1 Two-Pore-Domain Potassium (K_2P_) Channels

**DOI:** 10.3390/ijms20205193

**Published:** 2019-10-20

**Authors:** Felix Wiedmann, Daniel Schlund, Francisco Faustino, Manuel Kraft, Antonius Ratte, Dierk Thomas, Hugo A. Katus, Constanze Schmidt

**Affiliations:** 1Department of Cardiology, University of Heidelberg, 69120 Heidelberg, Germany; Felix.Wiedmann@med.uni-heidelberg.de (F.W.);; 2DZHK (German Center for Cardiovascular Research), Partner Site Heidelberg/Mannheim, University of Heidelberg, 69120 Heidelberg, Germany; 3HCR (Heidelberg Center for Heart Rhythm Disorders), University of Heidelberg, 69120 Heidelberg, Germany

**Keywords:** ion channel, K_2P_2.1, KCNK2, membrane trafficking, *N*-glycosylation, TREK-1, two-pore-domain potassium channels

## Abstract

Mechanosensitive hTREK-1 two-pore-domain potassium (hK_2P_2.1) channels give rise to background currents that control cellular excitability. Recently, TREK-1 currents have been linked to the regulation of cardiac rhythm as well as to hypertrophy and fibrosis. Even though the pharmacological and biophysical characteristics of hTREK-1 channels have been widely studied, relatively little is known about their posttranslational modifications. This study aimed to evaluate whether hTREK-1 channels are *N*-glycosylated and whether glycosylation may affect channel functionality. Following pharmacological inhibition of *N*-glycosylation, enzymatic digestion or mutagenesis, immunoblots of *Xenopus laevis* oocytes and HEK-293T cell lysates were used to assess electrophoretic mobility. Two-electrode voltage clamp measurements were employed to study channel function. TREK-1 channel subunits undergo *N*-glycosylation at asparagine residues 110 and 134. The presence of sugar moieties at these two sites increases channel function. Detection of glycosylation-deficient mutant channels in surface fractions and recordings of macroscopic potassium currents mediated by these subunits demonstrated that nonglycosylated hTREK-1 channel subunits are able to reach the cell surface in general but with seemingly reduced efficiency compared to glycosylated subunits. These findings extend our understanding of the regulation of hTREK-1 currents by posttranslational modifications and provide novel insights into how altered ion channel glycosylation may promote arrhythmogenesis.

## 1. Introduction

Two-pore-domain potassium (K_2P_) channels are implicated in the modulation of cellular excitability [1]. The 15 members of the K_2P_ family are characterized by a unique structure of two pore-forming loop domains in each subunit surrounded by four transmembrane helices [2]. K_2P_ subunits assemble as homo- or heterodimers to form functional background or “leak” potassium channels. Tandem of P domains in a weak inward rectifying K^+^ channel (TWIK)-related K^+^ channel 1 (TREK-1) channels are regulated by a variety of different physiological stimuli, such as membrane stretch, temperature, pH level, lipids, neurotransmitters, small molecules or signaling pathways [3]. The expression of TREK-1 channels has been reported in neuronal and cardiac tissues, as well as in the adrenal gland. TREK-1 channels have been proposed to be involved in multiple cellular and pathophysiological processes, such as processes associated with anesthesia, epilepsy, depression, neuroprotection in ischemia, sensation of pain, and regulation of cardiac rhythm as well as hypertrophy and fibrosis [4,5,6,7,8,9,10,11]. A heterozygous point mutation within the selectivity filter of hTREK-1 resulting in altered stretch-sensation and increased sodium permeability was recently identified in a patient with idiopathic right ventricular outflow tachycardia [12,13]. Furthermore, dysregulation of cardiac TREK-1 expression has been observed in patients suffering from atrial fibrillation and heart failure [14,15]. Even though the pharmacological [16,17,18,19,20,21,22] and biophysical [23,24,25,26,27,28] characteristics of hTREK-1 channels have been widely studied, relatively little is known about its posttranslational modifications [29].

Glycosylation of membrane proteins is a common co- or posttranslational modification. The addition of oligosaccharides is initiated in the endoplasmic reticulum, and the glycan is subjected to further alterations by rearrangement and trimming as the glycoprotein is trafficked through the Golgi apparatus to its destination at the cell membrane. *N*-linked glycosylation, facilitated by the addition of sugar moieties to the amino group of asparagine, constitutes one of the most common forms of glycosylation [30]. Unlike transcription or translation, glycosylation is entirely mediated in a nontemplated fashion. Therefore, the composition of the resulting glycan structure is highly dependent on the presence and activity of >200 glucosidases, glycosyltransferases, and transport proteins as well as their substrates. These molecules, however, vary between different types of cells and among cellular subcompartments and may change depending upon the metabolic state of a cell [31]. Thus, differential glycosylation of ion channels might represent a further process used by cells to increase functional diversity [32].

Similar to the case for other glycosylated membrane proteins, the quantity and composition of attached sugar moieties regulates the density and stability of many ion channels on the plasma membrane [33,34]. Furthermore, in several voltage-gated ion channels, negatively charged sialic acid residues located at the glycan structures are thought to contribute to the external negative surface potential, thereby impacting channel gating through electrostatic mechanisms [35,36,37]. While *N*-linked carbohydrate modifications have been described to alter the functional characteristics of the K_2P_ channels hK_2P_3.1 (hTASK-1, TWIK-related acid-sensing K^+^ channel 1), hK_2P_9.1 (hTASK-3, TASK-3, TWIK-related acid-sensing K^+^ channel 3) and hK_2P_18.1 (TRESK, TWIK-related spinal cord K^+^ channel) [38,39], it remains uncertain whether other members of the K_2P_ channel family are subjected to *N*-glycosylation in a similar fashion.

The aim of this work was to systematically assess the presence and functional implications of hTREK-1 channel glycosylation.

## 2. Results

Based on the observation that hTREK-1 channel subunits display electrophoretic mobility that accounts for a molecular weight higher than their calculated mass of 47.1 kDa, we wondered whether these subunits might be subjected to *N*-glycosylation. Bioinformatic prediction using the NetNGlyc 1.0 server (http://www.cbs.dtu.dk/services/NetNGlyc/) revealed two putative *N*-glycosylation consensus motifs at positions 110 and 134 with probability scores of 0.71 and 0.56, respectively (Figure 1a). Both sites, which consist of N-x-S/T motifs in which x can be any amino acid except proline, are localized at the extracellular M1-P1 interdomain of the hTREK-1 monomer (Figure 1b).

Two additional identified motifs were localized at intracellular domains that are not accessible for *N*-glycosylation. A three-dimensional model of the hTREK-1 channel dimer based on its recently revealed crystal structure [40,41,42] reveals that asparagine 110 is located at the summit of the overhead domain, while asparagine 134 is located at a more lateral position at the junction of the P1 loop (Figure 1c). The degree of conservation among vertebrate species is indicative of the potential functional relevance of these sites (Figure 1d). To explore whether hTREK-1 channels undergo *N*-linked carbohydrate modification, channel subunits equipped with C-terminal 1d4 immunolabels were heterologously expressed in *Xenopus laevis* oocytes, and protein lysates were subjected to gel electrophoresis followed by Western blotting (Figure 1e). Lysates of uninjected *Xenopus laevis* oocytes were included to determine the specificity of the 1d4 antibodies used. Upon coadministration of the antibiotic *N*-glycan synthesis inhibitor tunicamycin, a clear increase in electrophoretic mobility was observed.

To evaluate the effects of *N*-linked carbohydrate modification on the functional properties of hTREK-1 channels, *Xenopus laevis* oocytes heterologously expressing wild-type (WT) channel subunits were subjected to two-electrode voltage clamp (TEVC) measurements 24 and 48 h after injection. Due to the previous observation that the yolk sac of *Xenopus laevis* oocytes can sequester lipophilic compounds [19] two different routes of tunicamycin administration were chosen. Oocytes were either maintained in medium containing tunicamycin (incubated with 0.75 mg/L or 1.5 mg/L directly after RNA injection until TEVC measurements were performed) or coinjected with hTREK-1 RNA and 2 ng or 3 ng of tunicamycin per oocyte, respectively. Equal amounts of the vehicle, DMSO, were applied to the control oocytes.

Outward potassium currents evoked by a 500 ms test pulse from −80 mV to +20 mV are depicted in Figure 2a. The current amplitudes in oocytes administered tunicamycin were substantially lower than those of vehicle controls. The mean current amplitudes of *Xenopus laevis* oocytes after incubation or injection of tunicamycin or under the respective control conditions are provided in Figure 2b. Reductions in outward potassium currents were associated with marked depolarization of resting membrane potentials (RMPs). Upon inhibition of *N*-glycosylation by tunicamycin, the RMPs were shifted towards more positive values as displayed in the bottom part of Figure 2b. To study the implications of *N*-glycosylation on the current-voltage relationship of hTREK-1 currents, TEVC measurements were conducted, using the pulse step protocol, depicted in Figure 2d. From a holding potential of −80 mV, voltage steps (500 ms) were applied to potentials ranging from −80 to +60 mV in 20 mV increments at a frequency of 0.2 Hz (Figure 2c). Current-voltage relationships, investigated under isochronal recording conditions are visualized in Figure 2d, in which the currents under control conditions and after administration of tunicamycin are plotted as a function of the test pulse potential. While a statistically significant reduction in absolute currents was observed, the current-voltage relationships remained unchanged. Taken together, these data support the conclusion that hTREK-1 channel subunits undergo *N*-glycosylation and that inhibition of *N*-glycosylation is associated with reductions in potassium currents.

To gain further insight into the roles of the asparagine residues N110 and N134 as potential *N* glycan acceptors, two sets of channel mutant constructs were generated. As the presence of a proline residue in position +1 or +3 relative to the glycosylated asparagine has been reported to disrupt glycosylation motifs, we chose to substitute the +3 amino acid residues of both potential consensus sites with proline [43]. This was accomplished by generating the hTREK-1 mutant constructs E113P and N137P as well as E113P, N137P combining both mutations (Figure 3a,c,e,g,i). In a second approach, the predicted glycosylation sites were abolished by substitution of asparagine with glutamine, creating the mutant channels N110Q and N134Q as well as a N110Q/N134Q double-mutant channel (Figure 3b,d,f,h,j).

Even though introduction of the E113P mutation resulted in only partial disruption of the N110 glycosylation site (Figure 3a), the results of the proline mutant strategy were nearly identical to those obtained after glutamine substitution of N110 and N134. In monoglycosylated TREK-1 isoforms a second band, migrating at a slightly higher molecular weight can be observed that is completely absent in diglycosylated and nonglycosylated channel monomers (Figure 3a,b).

The mean current amplitudes of *Xenopus laevis* oocytes 24 or 48 h after injection of RNA encoding WT channels and glycosylation-deficient mutants are presented in Figure 3e,f. Current amplitudes of mutant constructs lacking either one or both *N*-glycosylation sites were substantially lower than those of WT channels. At the 48 h time point outward potassium currents of E113P constructs yielded 59 ± 2.4 % of their respective WT controls and N1137P constructs showed 42 ± 1.8% current, compared to their WT counterparts. In a similar fashion N110Q constructs displayed 67 ± 1.7 % and 134Q mutants showed 51 ± 1.1 % of their respective WT controls 48 h after RNA injection. As abolition of *N*-glycosylation sites resulted in decreased hTREK-1 current amplitudes, the RMPs were shifted towards more positive levels in a similar fashion (Figure 3e,f bottom part). For assessment of biophysical characteristics, outward potassium currents were elicited with the depicted pulse step protocol (Figure 3g,h). When the current amplitudes were plotted as a function of the test pulse potential, the glycosylation-deficient mutant constructs displayed current-voltage relationships virtually indistinguishable from those of their wild-type analogs (Figure 3i,j). Based on the comparable results of both mutagenesis strategies and the only partial disruption of *N*-glycosylation in the E113P mutant, the set of asparagine-to-glutamine mutants was chosen for further studies. To further assess whether basic biophysical and pharmacological properties of TREK-1 might be influenced by alterations in channel glycosylation state, WT and glycosylation-deficient TREK-1 channel subunits were exposed to different pH conditions as well as pharmacological TREK-1 modulators. The pH dependency of WT and glycosylation-deficient TREK-1 channels did not show statistically significant differences (Appendix A). The beta blocker carvedilol, recently reported to inhibit TREK currents [19], blocked WT and glycosylation-deficient TREK-1 channel subunits in a similar fashion (Appendix A). Finally, the small molecule TREK-1 current activator did not display significant differences on WT or glycosylation-deficient TREK-1 channels (Appendix A).

Consistent results were found upon heterologous expression of hTREK-1 channel subunits in mammalian cells, in the absence and presence of tunicamycin and after treatment of cell lysates with the *N*-glycosidase PNGase F (Figure 4a). When comparing the WT channels to the glycosylation-deficient mutant constructs N110Q and N134Q and the N110Q/N134Q double-mutant, three distinguishable bands were detected corresponding to diglycosylated, monoglycosylated and nonglycosylated channel subunits (Figure 4b). Figure 4c demonstrates that the WT TREK-1 channel subunits treated with PNGase F displayed the same apparent molecular weights as the N110Q/N134Q double-mutant constructs, confirming our hypothesis that asparagine residues 110 and 134 are the only active sites at which hTREK-1 channels are accessible for *N*-linked carbohydrate modification.

The observed reductions in the current amplitudes of nonglycosylated hTREK-1 channel subunits may have resulted from either altered single channel properties or reduced numbers of channels on the plasma membrane. To elucidate the cause, the cellular localization and cell surface trafficking of *N*-glycosylation-deficient channel constructs were further investigated by immunofluorescence microscopy after transient expression of channel mutant constructs in HeLa cells. Figure 5a confirms the correct surface trafficking of hTREK-1-WT-eGFP as well as the hTREK-1-WT-N110Q-eGFP, hTREK-1-N134Q-eGFP and hTREK-1-N110Q/N134Q-eGFP channel constructs. Furthermore, surface biotinylation assays were employed to quantitate the membrane trafficking of the *N*-glycosylation-deficient channel mutants. Representative immunoblots obtained after biotinylation of surface proteins and precipitation with avidin-coupled beads (right side, surface fraction) and of corresponding input lysates (left side, input) are presented in Figure 5b.

Both, monoglycosylated and nonglycosylated hTREK-1 subunits were detected in the surface fractions. A modest but consistent reduction in channel cell surface expression was, however, observed in the mutant constructs N110Q and N134Q, lacking one *N*-glycosylation motif (Figure 5c). After complete abolition of *N*-glycosylation, a significant decrease in cell surface expression was observed (*n* = 3 independent experiments; *p* = 0.015). While the current decline of monoglycosylated channel subunits was significantly higher (N110Q: 33.9 ± 1.7 %; N134Q: 49.4 ± 1.1%) than the observed reduction of surface protein expression (about 16 % for both, N110Q and N134Q), a current reduction of 65.5 ± 0.7 % in nonglycosylated subunits was accompanied by a reduction in surface expression of 70.2 %. Of note, tunicamycin treatment (incubation with 1.5 mg/L) resulted in a current reduction of 79.4%. Thus, the reduced current amplitudes of the *N*-glycosylation mutants may have arisen from inappropriate channel trafficking to the plasma membrane. 

Finally, to assess whether hTREK-1 channel subunits might be subjected to *O*-glycosylation, plasmid DNA encoding WT and completely *N*-glycosylation-deficient N110Q/N134Q channel subunits was transiently transfected into HEK-293T cells. Subsequently, the cells were treated with the *O*-glycosylation inhibitor benzyl 2-acetamido-2-deoxy-α-d-galactopyranoside (BenGal), which acts as a mimic of GalNAc-α-1-*O*-serine/threonine and thus completely inhibits *O*-glycan chain extension by blocking the β-1,3-galactosyltransferase involved in elongation of *O*-glycosides [44,45,46]. In a subset of samples, protein lysates were subjected to digestion with *O*-glycosidase. To yield *O*-glycosidic bonds accessible to enzymatic digestion, pretreatment with neuraminidase was necessary.As demonstrated in Figure 6, no changes in electrophoretic mobility could be observed upon *O*-glycosidase digestion or administration of the *O*-glycosylation inhibitor BenGal, suggesting the lack of a relevant role of *O*-glycosylation in posttranslational modification of hTREK-1 channels, at least in this heterologous expression system with HEK-239T cells.

## 3. Discussion

With the exception of hK_2P_15.1 (hTASK-5), a channel with unclear functional relevance, all members of the K_2P_ channel family carry at least one putative *N*-glycosylation motif in their M1-P1 interdomain [38]. While *N*-glycosylation of the channels hK_2P_1.1 (hTWIK-1), K_2P_3.1 (TASK-1), K_2P_9.1 (TASK-3) and K_2P_18.1 (TRESK) has been experimentally validated [38,39,47,48], relatively little is known about the glycosylation of the remaining K_2P_ channels. Functional validation of putative *N*-glycosylation motifs, however, is indispensable as Egenberger et al. [39] showed that TRESK harbors two putative *N*-glycosylation motifs, but only one is of functional relevance.

In our present work we show that hTREK-1 channel subunits undergo *N*-glycosylation at asparagine residues 110 and 134 and that the presence of sugar moieties at these two sites increases channel function. To ensure adequate administration of the lipophilic 844 kDa compound tunicamycin in *Xenopus oocytes*, two different routes of tunicamycin administration were applied. While in our study both strategies resulted in decreased TREK-1 currents, current suppression was higher in the incubation group, pointing towards a higher potency of tunicamycin in preventing TREK-1 glycosylation when applied in culture media. Further, two different mutagenesis strategies were used to abolish *N*-glycosylation motifs in the hTREK-1 channel sequence. Interestingly, even though introduction of the E113P mutation resulted in only partial disruption of the N110 glycosylation site, the proline mutants (E113P, N137P and E113P/N137P) displayed an even more pronounced current reduction compared to asparagine-to-glutamine mutants (N110 Q, N134Q and N110Q/N134Q).

24 h after RNA injection WT currents reached 57 % to 66 % of their full currents measured at the 48 h time point. Asparagine-to-glutamine mutants reached 54 % (N110Q), 41 % (N134Q) and 31 % (N110Q/N134Q) of their full currents at the 24 h time point, again reflecting a delay in surface trafficking of the glycosylation-deficient channel constructs. Respective currents of the proline mutants were as low as 8.8 % (E113P), 20.8 % (N137P) and 19.4 % (E113P/N137P). This observation points towards an additional delay in channel processing and trafficking in this set of mutants that could theoretically be attributed to the proline residues, introduced into the helices of the cap domain and the other loops potentially interfering with proper protein folding.

In direct comparison to N134Q, N110Q constructs display slightly enhanced migration properties in gel electrophoreses (Figure 3b, Figure 4b and Figure 5b). Therefore, one could speculate about whether the size and composition of the glucan attached to asparagine 110 might differ from the glucan attached to the asparagine in position 134. Further, abolition of the N134 glycosylation site either via introduction of the N134Q or the N137P mutation resulted in a more pronounced current decline as compared to deactivation of N110 glycosylation (i.e., the N110Q and E113P constructs). These results account for a higher functional relevance of the position 134 glycosylation site. Of note, surface biotinylation experiments did not show significant differences in surface expression of N110Q and N134Q constructs. It remains speculative whether these results might point towards an additional, trafficking-independent role of the N134 attached glucan in regulation of TREK-1 current.

A second band, migrating at a slightly higher molecular weight was observed in monoglycosylated TREK-1 subunits. This band could not be observed in both diglycosylated and nonglycosylated channel monomers (Figure 3a,b and Figure 4c). Based on the fact that it could not be observed in surface fractions (Figure 5b) one could speculate that band resembles immature channels subunits with incomplete processing and trimming of the attached sugar moieties.

Glycosylation at both sites was observed in *Xenopus laevis* oocytes as well as in mammalian cells and there was no evidence of further *N*-glycosylation sites or of *O*-glycosylation. Detection of glycosylation-deficient mutant channels in surface fractions and recordings of macroscopic potassium currents mediated by these subunits demonstrated that nonglycosylated hTREK-1 channel subunits can reach the cell surface in general, but with seemingly altered efficiency compared to glycosylated subunits. Thus, covalent addition of oligosaccharides at asparagine 110 or 134 increases the macroscopic current amplitude of hTREK-1 channel subunits by affecting their subcellular localization without significantly altering their pH sensitivity or pharmacological activation by ML335 and inhibition by carvedilol.

More than 20 years ago Lesage et al. [47] described the glycosylation of hTWIK-1 channels using glycosidase treatment and SDS-PAGE. The implications of such glycosylation on functional properties and protein trafficking, however, have not yet been studied. Likewise, the TASK-1 and TASK-3 channel sequences each harbor one *N*-glycosylation site in their P1-M1 interdomain, and functional glycosylation in heterologous expression systems has been revealed by Mant et al. [38]. This initial study reported a clear reduction in TASK-1 currents upon asparagine-to-glutamine mutation of the *N*-glycan acceptor site that was accompanied by reduced surface trafficking [38]. Goldstein et al. [48], however, demonstrated that proline substitution at the +3 position led to efficient prohibition of *N*-glycosylation without reductions in channel current or surface expression. It remains to be elucidated whether these differences arose from the use of different expression systems as Goldstein et al. used *Xenopus laevis* oocytes and Mant et al. [38] worked with HEK-239 cells. In closely related TASK-3 channels, deactivation of *N*-glycosylation led to slight reductions in surface expression without significantly changing whole-cell currents [38]. Taken together, these findings suggest that *N*-glycosylation has less fewer functional implications in members of the TASK-1 subfamily than in hTREK-1. As described by Egenberger et al. [39], TRESK channels also bear a single functional *N*-glycosylation site. Upon tunicamycin treatment or molecular biological abolition of glycan attachment, their surface expression is significantly reduced, resulting in decreased macroscopic currents, similar to the results obtained in our study [39]. Interestingly, one further putative *N*-glycosylation motif in the P1-M1 interdomain of TRESK that is not conserved in its murine ortholog has been shown to exhibit no functional relevance. Whether the observed reduction in channel abundance at the cell surface might have been caused by decreased surface trafficking of hTREK-1 channels, alterations in protein stability at the cell membrane, channel recycling or enhanced protein degradation warrants further investigation. Finally, modification of glycan structures by extracellular enzymes, such as sialidases, represents a mechanism that connects cellular functions to extracellular stimuli. Conversely, glycan structures can protect extracellular ion channel domains from cleavage by proteases and therefore allow cells to actively regulate responses to extracellular stimuli [49].

When introducing point mutations into the M1-P1 interdomain, care must be taken not to interfere with gating domains or channel dimerization. Therefore, results obtained from glycosylation-deficient mutants have to be interpreted with extreme caution. To this end, two different mutagenesis approaches were employed in this study, and the results were reproduced in experiments with the glycosylation inhibitor tunicamycin. It has further been reported that proteins may exhibit differences in *N*-glycosylation when expressed in heterologous cell systems compared to native tissue. The lack of specific TREK-1 inhibitors, however, currently prevents functional assessment of hTREK-1 currents in native tissue [10]. To reduce confounding factors regarding the expression systems, *Xenopus laevis* oocytes and two different mammalian cell lines were used in this study.

To date, more than 30 congenital disorders of glycosylation (CDGs) have been identified to alter glycan synthesis, modification or targeting [49]. As nearly all organ systems can be involved, CDGs display heterogeneous clinical phenotypes often accompanied by cardiomyopathies and arrhythmias that add to the significantly increased infant mortality rates associated with these disorders [50,51]. Reduction of repolarizing potassium currents results in action potential duration prolongation that can produce torsades de pointes tachycardia via dispersion of ventricular repolarization and promotion of early after-depolarizations [52]. To which extend dysregulated glycosylation of cardiac ion channels such as hTREK-1 might contribute to arrhythmogenesis in CDG remains to be elucidated.

As many cardiac ion channel subunits are glycosylated, hereditary mutations in the glycosylation motifs of hERG and KCNE1 channels have been reported to be causative of long-QT syndrome [53,54]. Decher et al. [13] recently reported that a heterozygous point mutation within the selectivity filter of hTREK-1 leading to altered stretch-sensation and increased sodium permeability was identified in a patient with idiopathic right ventricular outflow tachycardia [12,13]. Thus, slight changes in hTREK-1 current can theoretically result in detrimental alterations to cardiac electrophysiology.

In conclusion, in this report, we have shown that hTREK-1 channels are *N*-glycosylated at asparagine residues 110 and 134. Inhibition of hTREK-1 glycosylation through either tunicamycin treatment or disruption of glycosylation consensus motifs by introduction of different point mutations reduces potassium currents. As we demonstrated, this effect is mainly due to reduced surface targeting of hTREK-1 channels, while the current-voltage relationship remains unchanged.

These findings extend our understanding of the regulation of hTREK-1 currents by posttranslational modifications and provide novel insights into how altered ion channel glycosylation may promote arrhythmogenesis.

## 4. Materials and Methods

### 4.1. Molecular Biology

The 426 amino acid isoform of hTREK-1 (GenBank accession number EF165334) was subcloned into pMAX, a *Xenopus laevis* and mammalian cell dual purpose expression vector as reported previously [9]. Introduction of C-terminal proline linkers (RVPDGDPD) followed by 1d4 epitope tags (RVPDGDPDETSQVAPA) was performed using standard PCR techniques as described previously [55]. For generation of fluorescent protein reporter constructs, the enhanced Green Fluorescent Protein (eGFP) sequence was amplified from pEGFP-C2 (Clontech Laboratories, Mountain View, CA, USA) and subcloned into the C-terminus of hTREK-1. In all constructs used in this study, the open reading frame was preceded by an optimized translation initiation sequence (GCC GCC ACC). Site-directed PCR mutagenesis was performed as described previously [56] and the sequences of the mutant constructs were confirmed by DNA sequencing (GATC-Biotechnology, Konstanz, Germany) of both strands. Following vector linearization, RNA was produced using T7 RNA polymerase and a mMessage mMachine kit (Thermo Fisher Scientific, Waltham, MA, USA). The transcript concentrations were quantified using a NanoDrop spectrophotometer (ND-1000, peqLab Biotechnologie GmbH, Erlangen, Germany) and RNA integrity was assessed via agarose gel electrophoresis.

### 4.2. Solutions and Drugs

The *N*-glycosylation inhibitor tunicamycin (Sigma Aldrich, Steinheim, Germany) and the *O*-glycosylation blocker benzyl 2-acetamido-2-deoxy-α-d-galactopyranoside (BenGal) (Sigma Aldrich, Steinheim, Germany) were dissolved in dimethylsulfoxide (DMSO) to create stock solutions of 300 mM and 1 mg/mL, respectively and the stocks were stored at 20 °C.

### 4.3. Cell Culture and Transfection

Human embryonic kidney (HEK-293T) cells and HeLa cells (ATCC, Wesel, Germany) were cultured in Dulbecco’s modified Eagle’s medium (DMEM; Thermo Fisher Scientific, Waltham, MA, USA) supplemented with 10% fetal bovine serum (Thermo Fisher Scientific, Waltham, MA, USA), 100 U/mL penicillin G sodium, and 100 µg/mL streptomycin sulfate in an atmosphere of 95% humidified air and 5% CO_2_ at 37 °C. Transient transfections were performed by incubating 80% confluent cells in T75 tissue culture flasks (Sarstedt, Nümbrecht, Germany) overnight with 815 µL of 300 mM NaCl, 635 µL of ddH2O, 180 µL of polyethylenimine (0.323 g/L, Polysciences, Warrington, PA, USA) and 22 µg of plasmid DNA. For immunofluorescence staining, HeLa cells were seeded on glass coverslips prior to performing transient transfection with Lipofectamine 3000 (Thermo Fisher Scientific, Waltham, MA, USA) according to the manufacturer’s instructions. When indicated, the cell culture media were supplemented with 1 µg/mL tunicamycin, 300 µM BenGal or an equal amount of the respective solvent (DMSO).

### 4.4. Animal Handling and Oocyte Preparation

Ovarian lobes were surgically removed from female *Xenopus laevis* frogs (Xenopus express, Vernassal, France) under tricaine anesthesia (1 g/L, pH 7.5). After surgery, the frogs were allowed to recover consciousness. Oocyte collection was alternated between the left and right ovaries and was, followed by a recovery period of at least 30 weeks. No more than four surgeries were performed on any individual animal. After the final surgery, the anesthetized frogs were sacrificed by decerebration and pithing. This investigation conforms with the Guide for the Care and Use of Laboratory Animals (NIH Publication 85-23) and with Directive 2010/63/EU of the European Parliament. Approval was granted by the local Animal Welfare Committee (Regierungspraesidium Karlsruhe, reference number G221/12, 24.02.2016). Following collagenase digestion to ease removal of the follicular layer, defolliculated stage V–VI oocytes were manually selected under a stereomicroscope (Stemi 2000; Carl Zeiss Microscopy GmbH, Oberkochen, Germany). Injection of RNA (1.5–5 ng; 46 nL/oocyte) was performed, using an automatic injector (Nanoject II, Drummond, Broomall, PA, USA). Where indicated, the oocytes were either coinjected with 2–3 ng of tunicamycin per cell or incubated in medium containing 0.75–1.5 µg/mL tunicamycin to inhibit *N*-glycosylation. The respective vehicle controls were performed by injected or incubated with equal amounts of the solvent DMSO.

### 4.5. Protein Isolation and Immunoblot Analysis

Thirty-six hours after transient transfection, HEK-293T cells were collected in radioimmunoprecipitation assay (RIPA) buffer (150 mM NaCl, 20 mM Tris-HCl, 1 mM ethylene-diamine-tetraacetic acid (EDTA), 0.5% NP-40, 0.5% sodium deoxycholate, 1 mM NaF, 1 mM Na3VO4) by scraping on ice, agitated for 30 min at 4 °C and clarified by centrifugation at 15,000× *g* for 30 min at 4 °C. Thirty-six hours after RNA injection, *Xenopus laevis* oocytes were homogenized using glass Teflon tissue homogenizers (Wheaton/DWK Life Sciences, Millville, NJ, USA). The proteins were solubilized for 1 h at 4 °C in oocyte lysis buffer (100 mM NaCl, 40 mM KCl, 20 mM HEPES, 1 mM EDTA, 10% glycerol, and 1% 3-[(3-cholamidopropyl)dimethylammonio]-1-propanesulfonate (CHAPS); pH 7.4). Repeated centrifugation steps were then performed at 5000× *g* (5 min, 4 °C) to clarify the supernatant followed by a final centrifugation step at 15,000× *g* (20 min, 4 °C). The protein concentration was determined using a bicinchoninic acid (BCA) protein assay (Thermo Fisher Scientific, Waltham, MA, USA). All buffer solutions used for protein handling were supplemented cOmplete Mini Protease Inhibitor Cocktail (Roche Diagnostics, Mannheim, Germany). Where indicated, glucosidase digestion was performed, using PNGase F or neuraminidase and *O*-glucosidase (New England Biolabs, Ipswich, MA, USA) according to the manufacturer’s instructions. Prior to gel electrophoresis, the protein samples were boiled for 5 min in Laemmli sample buffer (Bio-Rad, Hercules, CA, USA) supplemented with 150 mM dithiothreitol and 5% β-mercaptoethanol at 95 °C. Protein immunodetection was performed after SDS gel electrophoresis and Western blotting as described previously [57]. In brief, nitrocellulose membranes were developed by sequential exposure to blocking reagent (phosphate-buffered saline (PBS) with 5% dry milk, 3% bovine serum albumin [BSA], 0.1% TWEEN 20); primary antibodies directed against the 1d4 immunolabel (mouse monoclonal anti rhodopsin, 1:100; sc-57432, Santa Cruz Biotechnology, Heidelberg, Germany) or glyceraldehyde-3-phosphate dehydrogenase (GAPDH; mouse monoclonal, 1:40,000; G8140-11, US Biological, Swampscott, MA, USA) and appropriate HRP-conjugated m-IgGκ binding protein particles (m-IgGκ BP-HRP, 1:3000; sc-516102, Santa-Cruz Biotechnology, Heidelberg, Germany). The signals were visualized using an enhanced chemiluminescence assay (GE Healthcare, ECL Western Blotting Reagents, Buckinghamshire, UK) and quantification was performed under a FluorChem Q luminescence detector (Cell BioSciences, Palo Alto, Ca, USA). Original pictures of the immunoblots are presented in the online supplementary material (see Appendix A).

### 4.6. Protein Surface Biotinylation

For surface expression assays, HEK-293T cells were incubated with 0.5 mg/mL cell-impermeable, noncleavable sulfo-NHS-SS-Biotin (Pierce, Rockford, IL, USA) for 30 min at 4 °C. Following biotinylation of lysine-exposed surface proteins, the oocytes were rinsed three times with 4 mL of glycine (10 mM in PBS, pH 8.0) to quench unbound biotin, and the proteins were solubilized as described above. The biotin-labeled proteins were isolated from the crude protein fractions by incubation with streptavidin beads (Pierce, Rockford, IL, USA) for 2 h at 4 °C. Following repeated wash steps, the surface proteins were liberated by boiling at 95 °C for 5 min in Laemmli solution containing 150 µM DTT buffer and 5% β-mercaptoethanol.

### 4.7. Two-Electrode Voltage Clamp Electrophysiology

One to three days after RNA injection, two-electrode voltage clamp recordings were obtained using an OC-725c amplifier (Warner Instruments, Hamden, CT, USA) connected to a Digidata 1322A A/D converter (Axon Instruments, Foster City, CA, USA). *Xenopus laevis* oocytes were transferred to the measurement chamber and continuously superfused with a standard extracellular solution that contained 96 mM NaCl, 5 mM 4-(2-hydroxyethyl)-1-piperazineethanesulfonic acid (HEPES), 4 mM KCl, 1.1 mM CaCl2 and 1 mM MgCl2 (pH 7.4). Microelectrodes were manufactured from borosilicate glass capillaries (GB 100F-10, Science Products, Hofheim, Germany) using a micropipette puller (Flaming/Brown P-87, Sutter Instruments, Novato, CA, USA). A 3 mM K^+^ solution was used as the pipette solution, and the pipette resistances ranged from 0.5–1.5 MΩ. All measurements were performed at room temperature (20–22 °C). Only recordings with leak currents <10% were considered for data analysis. Therefore, no leak subtraction was performed. The data were sampled at 2 kHz and filtered at 1 kHz.

### 4.8. Immunofluorescence Staining and Fluorescence Microscopy

Twenty-four hours after transient transfection with eGFP reporter-coupled ion channel constructs, HeLa cells were fixed with 4% paraformaldehyde for 15 min at room temperature. After incubation with PBS/Alexa Fluor 594 conjugated wheat germ agglutinin (WGA, 1 µg/mL; W11262, Thermo Fisher Scientific, Waltham, MA, USA) and bisbenzimide (10 µg/mL), repeated wash steps were performed. Finally, coverslips were mounted with FluorSave antifade solution (Merck Chemicals GmbH, Darmstadt, Germany) and examined using an Olympus IX73 microscope (Olympus, Shinjuku, Tokyo, Japan).

### 4.9. Statistical Analysis, Quantification and Data Presentation

The pCLAMP9 software (Version 9, Axon Instruments, Foster City, CA, USA) was used for data acquisition and analysis. Prism 5 (GraphPad Software, La Jolla, CA, USA) software was used for statistical analysis. The protein signals were quantified using ImageJ 1.41 software (National Institute of Health, Bethesda, MD, USA). Visualizations of the hTREK-1 crystal structure (PDB ID: 4TWK, revealed by Pike et al. 201441 [42]) were generated using a PyMOL Molecular Graphics System, Version 1.8 (Schrödinger LLC, Cambridge, MA, USA). Multiple sequence alignment was conducted with the Clustal Omega algorithm [58]. Unless stated otherwise, the data are expressed as the mean ± standard error of the mean (SEM). Paired and unpaired Student’s *t* tests (two-tailed tests) were used for statistical comparisons, and *p* < 0.05 was considered to indicate statistical significance. For multiple comparisons, one-way ANOVA was used. If the hypothesis of equal means could be rejected at the 0.05 level, pairwise comparisons of groups were made, and the probability values were adjusted for multiple comparisons using the Bonferroni correction.

## Figures and Tables

**Figure 1 ijms-20-05193-f001:**
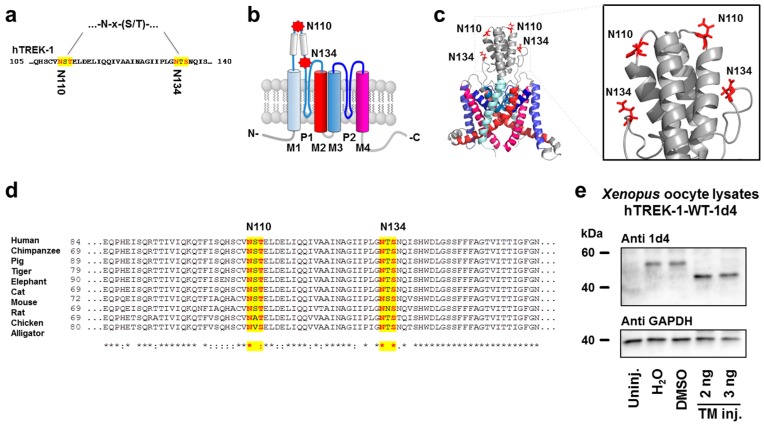
hTREK-1 channels harbor two putative *N*-glycosylation sites. (**a**) Two potential *N*-glycosylation consensus sites (yellow), consisting of an asparagine residue (N, red) followed by any amino acid except proline (x) and a serine or threonine residue (S/T, red) can be found in the extracellular part of the hTREK-1 amino acid sequence. Asparagine residues possibly modified by carbohydrates are located in positions 110 and 134. (**b**) Depicts a schematic membrane topology model of a hTREK-1 channel monomer consisting of two pore-forming loops (P1 and P2) surrounded by four transmembrane domains (M1–M4; top: extracellular space, bottom: intracellular space). Both potential *N*-glycosylation motifs are located in the extracellular M1-P1 interdomain. (**c**) Three-dimensional model of a hTREK-1 channel dimer, based on its crystal structure (PDB ID: 4TWK) [40,41,42]. N110 is located at the top of the overhead domain, and N134 is situated at a more lateral position. (**d**) A partial sequence alignment comparing hTREK-1 protein sequences of different species showing conservation of both motifs (‘*’, full conservation; ‘:’, conservative substitution; ‘.’, semiconservative substitution). (**e**) Immunoblot of hTREK-1-1d4 channel subunits heterologously expressed in *Xenopus laevis* oocytes. After coinjection of the antibiotic *N*-glycosylation inhibitor tunicamycin (TM), hTREK-1-1d4 proteins display increased electrophoretic mobility. The signals of glyceraldehyde 3-phosphate dehydrogenase (GAPDH) are provided as loading controls.

**Figure 2 ijms-20-05193-f002:**
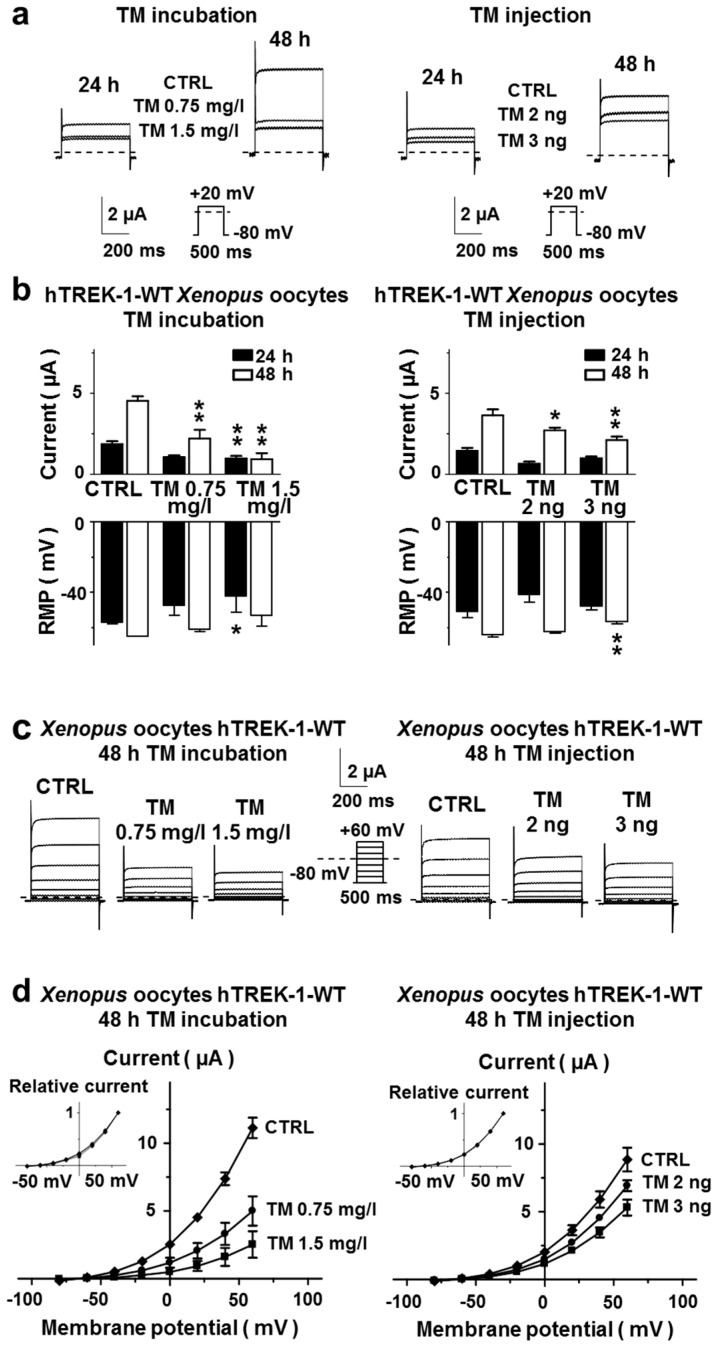
Inhibition of *N*-glycosylation by tunicamycin decreases TREK-1 currents. (**a**) The TREK-1 currents elicited by a 500 ms depolarizing voltage step from −80 mV to +20 mV (see depicted pulse protocol) are displayed under control conditions (CTRL) and 48 h after administration of TM by incubation (left) or cytoplasmic injection (right). Representative recording of *n* = 6–12 cells. (**b**) The mean potassium currents at +20 mV (top) and the resting membrane potentials (RMPs, bottom) are provided for different time points (24 h, black bars; 48 h, white bars) after TM incubation (left; *n* = 3–15) or cytoplasmic injection (right; *n* = 3–15 cells). (**c**) Representative families of macroscopic hTREK-1 potassium currents recorded from *Xenopus laevis* oocytes by application of the depicted pulse protocol. (**d**) Corresponding mean step current amplitudes plotted as functions of the test pulse potential showing comparable current-voltage relationships under control conditions and after application of TM. Inserts: the data are presented relative to the maximum current amplitude measured at +60 mV. The data are given as the mean ± standard error of the mean (SEM). The zero-current levels are indicated by dashed lines. The pulse protocols are depicted below the respective current traces. * *p* < 0.05, ** *p* < 0.01.

**Figure 3 ijms-20-05193-f003:**
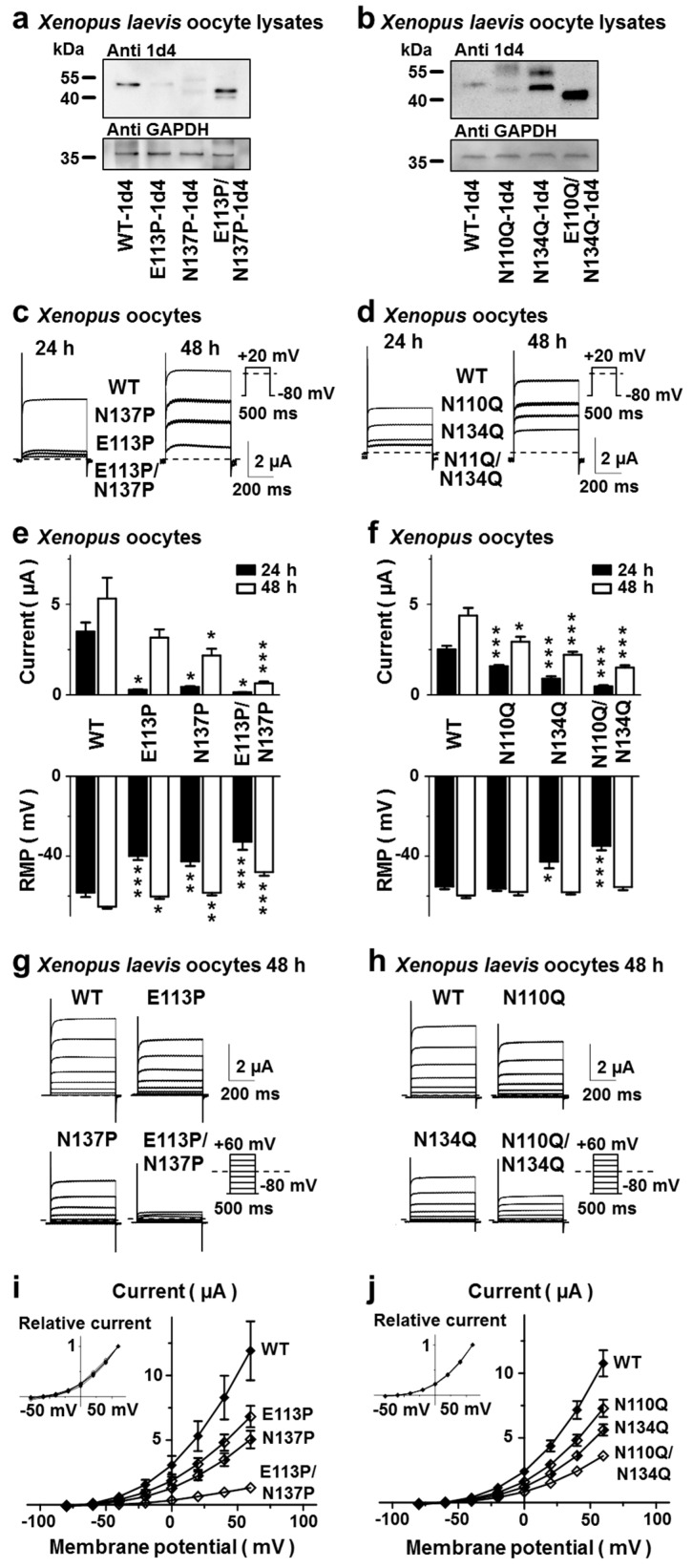
Molecular biological disruption of hTREK-1 *N*-glycosylation. (**a**) *N*-glycosylation can be disrupted either by introducing a proline residue in position +3 relative to the *N*-glycosylation acceptor residue (**a**,**c**,**e**,**g**,**i**) or by substituting the carbohydrate acceptor asparagine to glutamine (**b**,**d**,**f**,**h**,**j**). (**a**,**b**) Protein lysates of *Xenopus laevis* oocytes injected with RNA of the indicated hTREK-1-1d4 mutant constructs lacking the first, the second or both the first and second *N*-glycosylation motifs were subjected to immunoblotting. Changes in carbohydrate modifications are displayed as altered electrophoretic mobility. Please note that the E113P substitution resulted in incomplete disruption of *N*-glycosylation at N110. The immunosignals of glyceraldehyde 3-phosphate dehydrogenase (GAPDH) are given as loading controls. (**c**,**d**) Representative current traces recorded from *Xenopus laevis* oocytes expressing the indicated hTREK-1 variants after 24 h (left) or 48 h (right). The currents were evoked by the application of depolarizing voltage steps from −80 mV to +20 mV. Representative current traces of *n* = 4–12 cells are shown. The mean current amplitudes (top) and resting membrane potentials (bottom) of the cells included in this experiment are displayed in (**e**,**f**). (**g**,**h**) Families of hTREK-1 current traces evoked by the displayed pulse step protocols 48 h after injection of the hTREK-1 WT or mutant construct. (**i**,**j**) Activation curves recorded under isochronal conditions 48 h after RNA injection. Inserts: the data presented relative to the maximum current amplitude measured at +60 mV display comparable voltage-current relationships between glycosylated and nonglycosylated hTREK-1 channel subunits. The data are provided as the mean ± standard error of the mean (SEM). The dashed lines indicate the zero-current levels. The pulse protocols are depicted next to the current traces (* *p* < 0.05, ** *p* < 0.01, *** *p* < 0.001).

**Figure 4 ijms-20-05193-f004:**
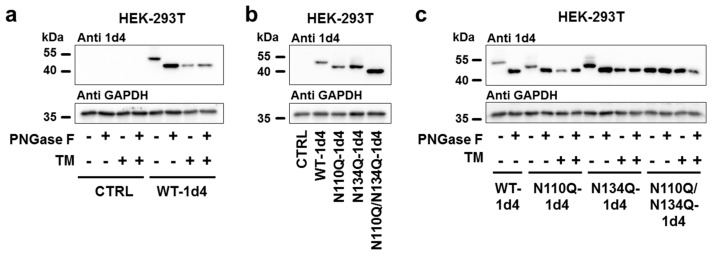
*N*-glycosylation of hTREK-1 currents, expressed in mammalian cells. To determine whether hTREK-1 channel subunits undergo *N*-glycosylation in mammalian cells, the relevance of the hTREK-1 *N*-glycosylation sites N110 and N134 was confirmed in HEK-293T cells. (**a**–**c**) Protein lysates of HEK-293T cells expressing hTREK-1-1d4 WT or mutant constructs were subjected to anti 1d4-immunoblotting under control conditions, after administration of tunicamycin (TM) or after cleavage of *N*-linked carbohydrates by the *N*-glycosidase PNGase F, as indicated by (+) or (−). Changes in carbohydrate modifications are displayed as altered electrophoretic mobility, and immunoblots of glyceraldehyde-3-phosphate dehydrogenase (GAPDH) are provided as loading controls.

**Figure 5 ijms-20-05193-f005:**
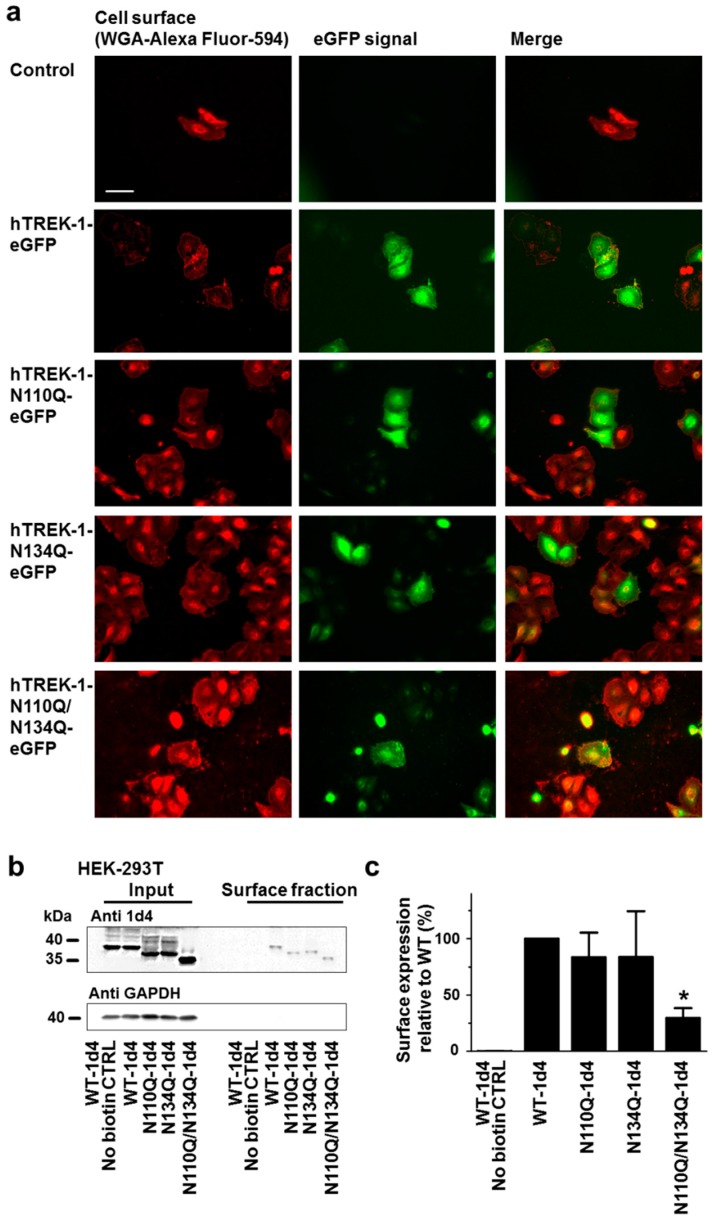
*N*-glycosylation regulates the surface expression of hTREK-1 channels. (**a**) EGFP-reporter coupled WT hTREK-1 channel subunits or glutamine mutants lacking either one or both *N*-glycosylation motifs were expressed in HeLa cells. Cell membranes stained with Alexa Fluor 594-labeled wheat germ agglutinin are depicted in red. The fluorescence signals of hTREK-1-eGFP variants are shown in green. The overlays (yellow) demonstrate colocalization of diglycosylated, monoglycosylated and nonglycosylated double-mutant channels with the cellular membrane, and show the preserved surface trafficking of deglycosylated channels. Scale bar: 10 µm. (**b**) Surface fractions of HEK-293T cells expressing the indicted hTREK-1 *N*-glycosylation-deficient variants were isolated via surface protein biotinylation, followed by streptavidin precipitation. Immunoblots of the input fractions are displayed on the left, and the mean immunosignals of the surface fractions are given on the right side (*n* = 3). (**c**) Ion channel subunit surface fractions (i.e., mean optical densities of the surface blots divided by the input fraction standardized by glyceraldehyde 3-phosphate dehydrogenase (GAPDH) as a loading control relative to the WT signal). The data are provided as the mean ± SEM (* *p* < 0.05).

**Figure 6 ijms-20-05193-f006:**
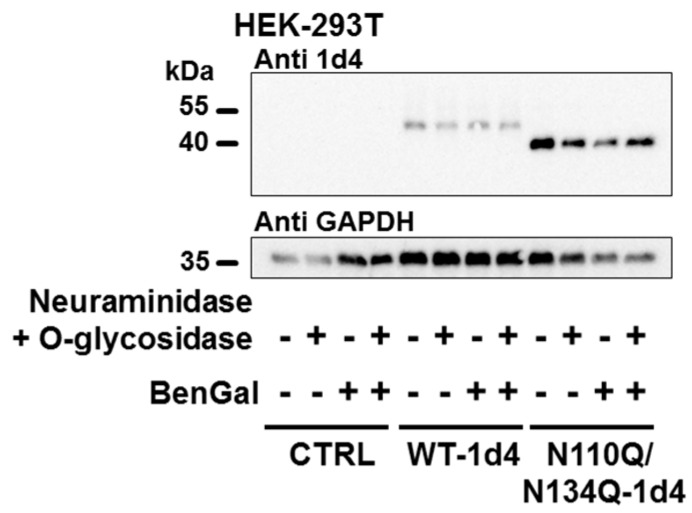
There is no evidence for *O*-glycosylation of hTREK-1 channels. Immunoblots of hTREK-1 are shown for control conditions, after administration of the *O*-glycosylation inhibitor benzyl 2-acetamido-2-deoxy-α-d-galactopyranoside (BenGal) and after incubation of the protein lysates with a mixture of *O*-glycosidase and neuraminidase (to yield potential *O*-glycosides accessible to *O*-glycosidase), as indicated by (+) and (−). However, no mobility shifts could be observed after treatment with BenGal or *O*-glycosidase, suggesting a lack of significant *O*-glycosylation of hTREK-1 in HEK-239T cells. The GAPDH signals are provided as loading controls.

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
