# Peer review of "N-Glycosylation of TREK-1/hK2P2.1 Two-Pore-Domain Potassium (K2P) Channels"

_ijms, 2019, doi:10.3390/ijms20205193_

Round 1

Reviewer 1 Report

The manuscript from Wiedmann et al. is an interesting work dealing with the role of N-glycosylation on the function of hTREK-1 channels.

The manuscript is properly written and the techniques used are appropriate to perform the study. There are many different control experiments that make the results very solid. Thus, the conclusions are in agreement with the results obtained. Therefore, this study is interesting and suitable for publication.

I have just some minor questions for the authors:

- In figure 2a, there is no Western blot for incubated TM. Is there any reason?

. Line 126, locates TM by incubation to the left and cytoplasmic injection to the right, but in the figure is put up and down.

- There is no reference in the text, neither in the result or the discussion section about the differences of the experiments performed with TM incubation or injection. If you put results for both experiments, it is expected some kind of discussion about them.

- In figure 3a, results from E113P and N137P are not clear. You mention a partial disruption of the N110 glycosylation site but for E113P I cannot see a band at a lower position. If fact the bands are very faint for both single mutants. However, for the double mutant the result is quite clear. Any comment about it?

- In figure 3b there are other bands above N110Q and N134Q. Any comment?

- In figure 3e, white bars at 48h for mutants seem to have significant differences with WT but there is no asterisk over them. Is it correct?

- Any comment about the results at 24 vs 48h?

- In figure 5d bands are not visible in the surface fraction.

- Have you considered looking for any numerical correlation between the fall in the surface fraction of the protein (fig. 5c) and that of the macroscopic current between the different mutants (fig. 3f)?

- line 291, “at asparagine 110, the proline mutants displayed…” Do you mean the glutamine mutants?

- Could you evaluate the effects of glycosylation of N110 and N134 separately? I mean, are both processes equivalent in the sense of diminishing the current or the surface location of the channel?

- Could you relate, even in a speculative way, the lower current, RPM and surface trafficking of hTREK with arrhythmogenesis?

Author Response

Reviewer 1:

The manuscript from Wiedmann et al. is an interesting work dealing with the role of N-glycosylation on the function of hTREK-1 channels.

The manuscript is properly written and the techniques used are appropriate to perform the study. There are many different control experiments that make the results very solid. Thus, the conclusions are in agreement with the results obtained. Therefore, this study is interesting and suitable for publication.

Thank you for reviewing the manuscript, for highlighting important implications of our work and providing positive feedback.

I have just some minor questions for the authors:

- In figure 2a, there is no Western blot for incubated TM. Is there any reason?

Immunoblot experiments presented in former Figure 2a were initially designed as an assay to screen for N-glycosylation of TREK-1 channels In vitro. Therefore the simplest way of tunicamycin administration (i.e. coinjection) was chosen. The results presented in former Figure 2a justified the way of tunicamycin-administration as the endpoint of complete deglycosylation on the western blot membrane was met. When in the context of a second approach functional implications of deglycosylation were studied using two-electrode voltage clamp measurements, the problem of insufficient cell viability and increasing leak currents under long term tunicamycin treatment was encountered. To detect the maximum achievable current reduction under tunicamycin treatment different ways of tunicamycin-administration were tested and finally the dataset presented in former Figure 2b-e was acquired. On that account in the revised version of our manuscript we decided to move Figure 2a to Figure 1 (as the new panel Figure 1e), the figure that is dealing with the question whether TREK-1 subunits might be subjected to N-glycosylation, leaving Figure 2 focused on the electrophysiological effects of tunicamycin treatment. We like to thank Reviewer 1 for helping us to improve clarity in data presentation.

If you however request blots with TM incubation we would be willing to perform the requested experiments in the next few weeks.

- Line 126, locates TM by incubation to the left and cytoplasmic injection to the right, but in the figure is put up and down.

We would like to thank reviewer 1 for pointing out this imprecision. Due to the panel rearrangements introduced during the revision (see Reviewer 1 point 1) the directions left and right now are appropriate.

- There is no reference in the text, neither in the result or the discussion section about the differences of the experiments performed with TM incubation or injection. If you put results for both experiments, it is expected some kind of discussion about them.

In the revised version our manuscript an explanation was added why tunicamycin was administered via injection and incubation: “Due to the previous observation that the yolk sac of Xenopus laevis oocytes can sequester lipophilic compounds [19] two different routes of tunicamycin administration were chosen.” The following paragraph was added to the discussion section: “To ensure adequate administration of the lipophilic 844 kDa compound tunicamycin in Xenopus oocytes two different routes of tunicamycin administration were applied. While in our study both strategies resulted in decreased TREK-1 currents, current suppression was higher in the incubation group, pointing towards a higher potency of tunicamycin in preventing TREK-1 glycosylation when applied in culture media.”

- In figure 3a, results from E113P and N137P are not clear. You mention a partial disruption of the N110 glycosylation site but for E113P I cannot see a band at a lower position. In fact the bands are very faint for both single mutants. However, for the double mutant the result is quite clear. Any comment about it?

Introduction of the N137P mutation completely abolished glycosylation at the N134 position while E113P resulted in only weak inhibition of N110 glycosylation. Interestingly, as Reviewer 1 mentioned correctly: Inhibition of N110 glycosylation by E113P was significantly higher in the double mutant construct, harboring E113P and N137P. Perhaps, either loss of the N134 glycan modification or conformational changes due to insertion of two proline residues in the M1-P1 interdomain instead of one contributed to partially inhibiting N110 glycosylation in the double mutant construct. As the asparagine to glutamine strategy managed to abolish N-glycosylation completely, glutamine constructs were used for further studies. We however decided to present the results of both mutagenesis strategies in our manuscript as they kind of replicated the results obtained with asparagine to glutamine mutants without “touching” the respective asparagine amino acid residues.

- In figure 3b there are other bands above N110Q and N134Q. Any comment?

Interestingly, the bands mentioned by Reviewer 1 can be observed only in monoglycosylated TREK-1 isoforms (N110Q and N134Q as well as N137P; in E113P the second band is missing but this could be explained due to the weak primary band). Figure 4c shows that this effect can be observed in mammalian cells in a similar fashion (even though the bands are less prominent) where they are not able to reach the cell surface. One could speculate whether these bands resemble immature channels subunits in which processing and trimming of the attached sugar moieties has not been completed. This theory would explain why the band is missing in nonglycosylated double mutant subunits but it is not entirely clear why this immature forms should be completely absent in WT channels. Maybe the lack of the second glycan (and the trafficking problems discussed in our manuscript) could be causative in retaining these incomplete processed subunits.

As all these considerations remain highly speculative and experimental validation would be beyond the scope of this project. Therefore we initially decided to not include this issue in the discussion section of our manuscript.

In the revised version of our manuscript the following paragraphs were added: Results: “In monoglycosylated TREK-1 isoforms a second band, migrating at a slightly higher molecular weight can be observed that is completely absent in diglycosylated and nonglycosylated channel monomers (Figure 3ab) “. Discussion: “A second band, migrating at a slightly higher molecular weight was observed in monoglycosylated TREK-1 subunits. This band could not be observed in both, diglycosylated and nonglycosylated channel monomers (Figure 3a,b and Figure 4c). Based on the fact that it could not be observed in surface fractions (Figure 5b) one could speculate that band resembles immature channels subunits with incomplete processing and trimming of the attached sugar moieties.”

- In figure 3e, white bars at 48h for mutants seem to have significant differences with WT but there is no asterisk over them. Is it correct?

We like to thank Reviewer 1 for pointing out the lack of asterisks in this particular group. This issue has now been corrected:

- Any comment about the results at 24 vs 48h?

Following your suggestion the Discussion part of the revised manuscript contains the following section: “24 h after RNA injection WT TREK-1 currents reached 57 % to 66 % of their full currents measured at the 48 h time point. Asparagine to glutamine mutants reached 54 %(N110Q), 41 % (N134Q) and 31 % (N110Q/N134Q) of their full currents at the 24 h time point, again reflecting a delay in surface trafficking of the glycosylation deficient channel constructs. Respective currents of the proline mutants were as low as 8.8 % (E113P), 20.8 % (N137P) and 19.4 % (E113P/N137P). This observation points towards an additional delay in channel processing and trafficking in this set of mutants that could theoretically be attributed to the proline residues, introduced into the helices of the cap domain and the other loops potentially interferring with proper protein folding.”

- In figure 5b bands are not visible in the surface fraction.

Thank you very much for pointing out this technical issue that now has been corrected. In the revised version of our manuscript Figure 5 is presented as follows:

- Have you considered looking for any numerical correlation between the fall in the surface fraction of the protein (fig. 5c) and that of the macroscopic current between the different mutants (fig. 3f)?

Following your suggestion the correlation of electrophysiological and proteinbiochemical results is discussed as follows in the revised version of our manuscript: “While the current decline of monoglycosylated channel subunits was markedly higher (N110Q: 33.9 ± 1.7 %; N134Q: 49.4 ± 1.1%) than the observed reduction of surface protein expression (about 16 % for N110Q as well as N134Q), a current reduction of 65.5 ± 0.7 % in nonglycosylated subunits was accompanied by a reduction in surface expression of 70.2 %. Of note, tunicamycin treatment (incubation with 1.5 mg/l) resulted in a current reduction of 79.4%.”

- line 291, “at asparagine 110, the proline mutants displayed…” Do you mean the glutamine mutants?

We agree with Reviewer 1 that the initial sentence: “Interestingly, even though introduction of the N113P mutant resulted in incomplete inhibition of glycan attachment at asparagine 110, the proline mutants displayed even more pronounced current reduction.” was a little bit confusing (Please see Figure 3e-f). Therefore, in the revised version of our manuscript the sentence was changed to “Interestingly, even though introduction of the E113P mutation resulted in only partial disruption of the N110 glycosylation site, the proline mutants (E113P, N137P and E113P/N137P) displayed an even more pronounced current reduction compared to asparagine to glutamine mutants (N110Q, N134Q and N110Q/N134Q).”. We would like to thank Reviewer 1 for helping us to improve clarity in our wording.

Figure 3e-f:

- Could you evaluate the effects of glycosylation of N110 and N134 separately? I mean, are both processes equivalent in the sense of diminishing the current or the surface location of the channel?

Thank you for raising this interesting issue. The Results section now states: “At the 48 h time point outward potassium currents of E113P constructs yielded 59 ± 2.4 % of their respective WT controls and N1137P constructs showed 42 ± 1.8 % current, compared to their WT counterparts. In a similar fashion N110Q constructs displayed 67 ± 1.7 % and 134Q mutants showed 51 ± 1.1 % of their respective WT controls 48 h after RNA injection.” The following paragraph was added to the discussion section: “In direct comparison to N134Q, N110Q constructs display slightly enhanced migration properties in PAGE (Figure 3b, Figure 4b and Figure 5b). Therefore, one could speculate about whether the size and composition of the glucan attached to asparagine 110 might differ from the glucan attached to the asparagine in position 134. Further, abolition of the N134 glycosylation site either via introduction of the N134Q or the N137P mutation resulted in a more pronounced current decline as compared to deactivation of N110 glycosylation (i.e. the N110Q and E113P constructs). These results account for a higher functional relevance of the position 134 glycosylation site. Of note, surface biotinylation experiments did, not show significant differences in surface expression of N110Q and N134Q constructs. It remains speculative whether these results might point towards an additional, trafficking-independent role of the N134 attached glucan in regulation of TREK-1 current.”

- Could you relate, even in a speculative way, the lower current, RPM and surface trafficking of hTREK with arrhythmogenesis?

The Discussion section of our revised manuscript now contains the following explanation: Reduction of repolarizing potassium currents results in action potential duration prolongation that can produce torsades de pointes tachycardia via dispersion of ventricular repolarization and promotion of early after-depolarizations [52]. Therefore, the citation “52. Tamargo, J.; Caballero, R.; Gómez, R.; Valenzuela, C.; Delpón, E. Pharmacology of cardiac potassium channels. Cardiovasc. Res. 2004, 62, 9–33. Doi: 10.1016/j.cardiores.2003.12.026.” was added.

Reviewer 2 Report

This manuscript describes the effect of glycosylation on hTREK1 channels.  The results suggest that glycosylation primarily affects trafficking of the channel to the plasma membrane.  Other measures of channel function do not appear to be affected in a significant manner.  The manuscript studies are well designed and the results are clear.  Well written, but spell check is suggested.  

Author Response

Reviewer 2:

This manuscript describes the effect of glycosylation on hTREK1 channels. The results suggest that glycosylation primarily affects trafficking of the channel to the plasma membrane.  Other measures of channel function do not appear to be affected in a significant manner.  The manuscript studies are well designed and the results are clear.  Well written, but spell check is suggested.

Thank you for reviewing the manuscript and for your very positive evaluation of our data. During the course of our revision the manuscript was subjected to a thorough spell check.